# High-quality Text-to-3D Character Generation with SparseCubes and Sparse Transformers

**Jiachen Qian[1], Hongye Yang[1], Shuang Wu[1,2], Jingxi Xu[1], Feihu Zhang[1]***
[1]DreamTech, [2]Nanjing University

## Abstract

Current state-of-the-art text-to-3D generation methods struggle to produce 3D models with fine details and delicate structures due to limitations in differentiable mesh representation techniques. This limitation is particularly pronounced in anime character generation, where intricate features such as fingers, hair, and facial details are crucial for capturing the essence of the characters.

In this paper, we introduce a novel, efficient, sparse differentiable mesh representation method, termed SparseCubes, alongside a sparse transformer network designed to generate high-quality 3D models. Our method significantly reduces computational requirements by over 95% and storage memory by 50%, enabling the creation of higher resolution meshes with enhanced details and delicate structures. We validate the effectiveness of our approach through its application to text-to-3D anime character generation, demonstrating its capability to accurately render subtle details and thin structures (*e.g.,* individual fingers) in both meshes and textures.

## 1 Introduction

Recent advancements in 3D generation have primarily concentrated on transforming textual descriptions into 3D models. Techniques such as DreamFusion (Poole et al., 2022) and subsequent methodologies employ score distillation sampling to iteratively refine an implicit Neural Radiance Field (NeRF) (Mildenhall et al., 2020) representation, facilitating the creation of detailed 3D models. In a similar vein, Tang et al. (2024b) introduce implicit 3D Gaussians for enhancing 3D model generation.

One-2-3-45++ (Liu et al., 2023a) and XCube(Ren et al., 2024) discretize 3D spaces into voxel grids and uses diffusion to learn and optimize 3D meshes. Similarly, DMTet (Shen et al., 2021) and Flexicubes (Shen et al., 2023) employ a grid-based method, but they advance it further by using densely packed cubes within the grid. Unlike traditional voxels, these cubes incorporate controllable geometric features such as Signed Distance Fields (SDFs) (Park et al., 2019), enabling DMTet and Flexicubes to generate smoother and more accurate mesh surfaces.

Recently, large reconstruction models (Hong et al., 2023; Wang et al., 2023b; Li et al., 2023a) have been introduced for single or multi-view based 3D reconstruction and generation. These models typically encode input images as tokens and train a powerful transformer architecture to reconstruct the objects using implicit representations like NeRF (Triplanes) (Li et al., 2023a; Hong et al., 2023), 3D Gaussians (Tang et al., 2024a) or DMTet/Flexicube (Peng et al., 2024; Xu et al., 2024).

However, a key limitation of these methods lies in their struggle to capture fine details and intricate structures. This is primarily due to the challenges associated with the differentiable representation of the meshes. Character generation exemplifies this issue, particularly with elements such as hands, hair, and facial features, which are critical for capturing a character's essence (as shown in Fig. 1). Unfortunately, compressing the 3D space into implicit representations, such as 2D Triplanes (Li et al., 2023a) (Fig. 1a (a)), 3D Gaussian (Tang et al., 2024a) (Fig. 1b) often leads to a significant number of floating-point artifacts on the recovered 3D meshes. To mitigate this issue, existing

---

*Corresponding author.

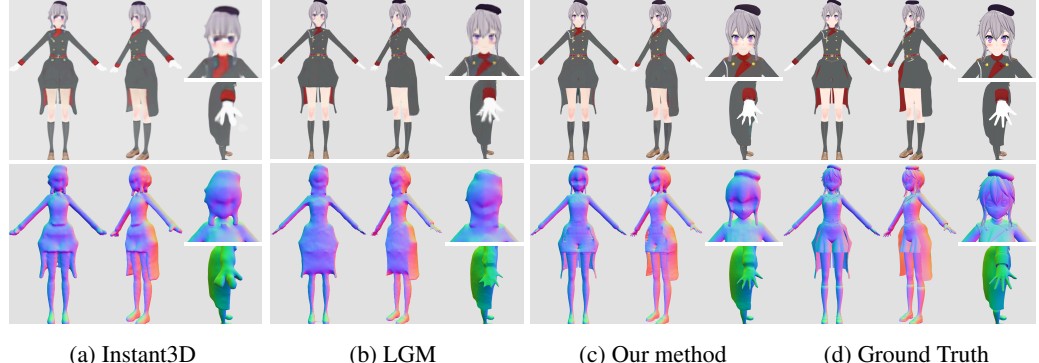

| (a) Instant3D | (b) LGM | (c) Our method | (d) Ground Truth |

Figure 1: Visualization of the generation results with other methods. All algorithms used the same set of multi-view images as input to generate the current avatar. (a) Instant3D (Li et al., 2023a) produces blurred textures and overly smoothed meshes. (b) LGM (Tang et al., 2024a) provides relatively clear rendered images but fails to provide detailed geometric structures. (c) Our method can generate high-quality results, producing clear textures and detailed geometric structures even in intricate areas such as the face and hands. (d) Ground truth.

methods often employ strong smoothing filters, which unfortunately remove crucial details in the process.

Voxel-based methods (Schwarz et al., 2022; Ren et al., 2024) offer a way to represent 3D space but often suffer from voxel effects on the final mesh surfaces (Fig. 6). While increasing the resolution can mitigate these artifacts, it comes at a significant cost. Memory usage and computational requirements grow cubically, making high-resolution approaches impractical. Methods such as DMTet and Flexicubes address this issue by incorporating Signed Distance Fields (SDFs) and geometric information into each cube, enabling the extraction of high-quality meshes. However, this approach also relies on a dense, high-resolution space, which leads to high memory and computational demands, especially when aiming for very detailed reconstructions or generations.

We propose a novel approach to address the challenges of high-resolution 3D mesh generation: the Differentiable SparseCubes representation. This representation enables the efficient learning of detailed 3D meshes and textures with significantly lower memory and computational costs compared to other methods. To leverage this new representation, we introduce a 3D Sparse Cube Transformer architecture specifically designed for processing SparseCubes. The combination of these innovations ensures superior preservation of geometric details, ultimately leading to the generation of high-quality meshes.

The effectiveness of our SparseCubes representation and Sparse Cube Transformers is demonstrated in the task of text-to-3D character generation. As illustrated in Fig. 1, our method excels in producing high-quality meshes and textures that feature exceptional details and intricate structures, particularly in crucial areas such as faces and hands.

## 2 RELATED WORK

### 2.1 3D REPRESENTATION

Previous methods (Wu et al., 2016; Gadelha et al., 2017; Henzler et al., 2019; Lunz et al., 2020; Smith & Meger, 2017) employ voxels to represent 3D objects. However, due to the high memory footprint of voxel-based representations, alternative 3D representations such as point clouds (Achlioptas et al., 2018; Yang et al., 2019; Zhou et al., 2021; Mo et al., 2019), occupancy networks (Mescheder et al., 2019), Signed Distance Fields (SDFs) (Park et al., 2019), and octrees (Ibing et al., 2023) have been explored. Recently, Neural Radiance Fields (NeRF) (Mildenhall et al., 2020) have emerged as a groundbreaking implicit 3D representation, widely used in 3D reconstruction (Zhang et al., 2020; Müller et al., 2022; Barron et al., 2022; 2023; Hu et al., 2023; Li et al., 2023b) and generation (Poole et al., 2022; Lin et al., 2023; Chan et al., 2022). Instead of purely using multi-layer perceptrons (MLPs) to implicitly represent the scene, TensoRF (Chen et al., 2022)

and Strivec (Gao et al., 2023) utilize tensor grids. Additionally, 3D Gaussian Splatting (3D GS) has been proposed as an alternative 3D representation to NeRF, demonstrating impressive quality and speed. Liao et al. (2018) apply the Marching Cubes (MC) algorithm to extract surface meshes from implicit representations. DMTet (Shen et al., 2021) represents a shape using a Signed Distance Field (SDF) encoded with a deformable tetrahedral grid and directly utilizes explicit supervision on the surface, thus generating high-quality shapes with arbitrary topology. FlexiCubes (Shen et al., 2023) utilize dense regular cubes to capture the geometry.

## 2.2 3D GENERATION

**Score Distillation Sampling (SDS) based Methods.**    As diffusion models have achieved remarkable results in the text-to-image task, there is an emerging trend of employing pretrained 2D text-to-image diffusion models to perform text-to-3D synthesis. Dreamfusion (Poole et al., 2022) introduces a score distillation sampling (SDS) loss, which transfers pretrained 2D image-text diffusion models to 3D object synthesis without any 3D data. Magic3D (Lin et al., 2023) combines the hash grid representation (Müller et al., 2022) with the image diffusion prior and refines the model in a coarse-to-fine manner. Zero-1-to-3 (Liu et al., 2023b) fine-tunes a pretrained diffusion model to learn controls over camera parameters and employs Score Jacobian Chaining (SJC) (Wang et al., 2023a) to optimize a 3D representation with priors from text-to-image diffusion models. Dream-Booth3D (Raj et al., 2023) first fine-tunes the DreamBooth (Ruiz et al., 2023) model and uses SDS loss to optimize the NeRF, then renders multi-view results from the NeRF and translates them into pseudo multi-view images. These images are used to fine-tune both the DreamBooth model and the NeRF. Make-it-3D (Tang et al., 2023) first applies SDS loss to leverage 2D diffusion priors to generate a coarse model with plausible geometry, then refines the texture of the model using both visible regions in the reference image and the diffusion prior. Realfusion (Melas-Kyriazi et al., 2023) uses single-image textual inversion as a substitute for alternative views and normal vector regularization to avoid low-level artifacts on surfaces. NeRDi (Deng et al., 2023) concatenates text embeddings from an image caption network and Textual Inversion (Gal et al., 2022), using the Pearson correlation as a geometric regularization term. Fantasia3D (Chen et al., 2023) disentangles geometry and appearance modeling and introduces the spatially-varying Bidirectional Reflectance Distribution Function. ProlificDreamer (Wang et al., 2023c) introduces Variational Score Distillation (VSD), exhibiting better generation results than SDS. Magic123 (Qian et al., 2024) incorporates both 2D and 3D priors into the 3D generation process and optimizes the model in a coarse-to-fine manner. DreamGaussian (Tang et al., 2024b) combines 3D Gaussian splatting (Kerbl et al., 2023) and SDS loss for efficient 3D content generation.

**Reconstruction based Methods.**    LRM (Hong et al., 2023) takes an image as input and adopts a large transformer-based encoder-decoder architecture for learning 3D representations of objects. PF-LRM (Wang et al., 2023b) employs a large transformer model to predict camera poses and reconstruct 3D objects from a few unposed images. DMV3D (Xu et al., 2023) uses LRM as the multi-view denoiser, aiming to predict a clean Triplane representation from noisy multi-view images. Instant3D (Li et al., 2023a) first generates four-view images simultaneously via a fine-tuned text-to-image diffusion model and then regresses a Triplane-based (Chan et al., 2022) NeRF from these images using a transformer-based model. CRM (Wang et al., 2024) designs a Convolutional Reconstruction Model and integrates geometric priors using six orthographic images. LGM (Tang et al., 2024a) combines 3D Gaussian splatting with a proposed asymmetric U-Net to facilitate high-resolution training. Our method utilizes a similar transformer network. While, we implement a more efficient SparseCubes representation and Sparse 3D transformer for high-quality 3D generation.

## 3 METHOD

To realize a text-to-3D pipeline, we first convert the text into multi-view images. Due to the limited amount of 3D character training data, we fine-tune a text-to-image model based on PIXART-$\Sigma$ (Chen et al., 2024). We then train a multi-view diffusion model similar to ImageDream (Wang & Shi, 2023), but with a higher resolution (512).

Our contribution lies in the proposed SparseCubes (Sec. 3.1) and Sparse Cube Transformer (Sec. 3.2) that can convert four image views into high-quality textured 3D character models. As illustrated

in Fig. 2, the proposed method contains two parts: 1) a low-resolution coarse mesh proposal network used to sparsify the cubes in the 3D space, and 2) a high-resolution Sparse Cube Transformer used to further extract accurate meshes and textures.

Since learning and optimization in 3D spaces require cubic computations and memory costs, existing methods like LRM, Instant3D, and InstantMesh *et al.*encode 2D features and optimize the Triplanes, and then interpolate to achieve the 3D object. The compression of 3D information to 2D sometimes leads to skew artifacts and over-smoothness of 3D meshes, as illustrated in Fig. 1 and Fig. 5. We instead utilize 3D transformers that can preserve the 3D geometric information and produce better meshes.

## 3.1 SPARSECUBES

Although current differentiable 3D representations, such as NeRF and Gaussian Splatting, can render high-quality RGB images, extracting high-quality meshes from them remains challenging.

Schwarz et al. (2022) and Ren et al. (2024) use 3D voxels to extract meshes but result in only coarse meshes with voxel effects on mesh surfaces (Fig. 6). Using high-resolution voxels can reduce these artifacts, but the memory and computational costs grow cubically. Similarly, DMTet (Shen et al., 2021) and FlexibleCube (Shen et al., 2023) can produce smooth meshes. However, limited by the 3D resolution, they cannot preserve details and delicate structures (*e.g.,*fingers).

To achieve high-quality meshes, we specifically design a new differentiable 3D representation, named SparseCubes. As illustrated in Fig. 2 (a), we implement both geometric features and texture features into one cube. For the geometric feature, similar to FlexiCubes (Shen et al., 2023), we use a 25-dimensional feature to represent the mesh. The first dimension represents the SDF of each vertex, while the second to fourth dimensions represent the deformation of each vertex. The remaining 21 dimensions consist of three sets of weights defined within each cube. These weights are used to adjust the location of the crossing point along each edge, the dual vertex inside each face, and the midpoint, respectively.

Different from other methods (Shen et al., 2021; 2023) that use a separate implicit representation to interpolate RGB texture information in order to reduce memory and computational costs, we directly implement a 30-dimensional feature vector into each cube to represent the texture information. This means both geometry and texture can be optimized simultaneously.

We observe that the vast majority ($> 95\%$) of cubes are empty, with only a small portion containing mesh faces or vertices. Therefore, we do not use dense cubes but instead select a small subset (less than $5\%$) to save memory and computational costs. The sparsification process is shown in Fig. 2 (b). We design a criterion to extract SparseCubes from these dense cubes. For a cube, if the SDF values of all eight corners of a cube are either greater than 0 or less than 0, it indicates that the cube is entirely outside or inside the mesh, respectively. To select cubes contain faces, we use the following criterion:

$$T > \sum_{i \in \omega} \mathbb{I}(\text{SDF}_i > 0) > 0, \tag{1}$$

where $\omega$ represents the set of all corners of the cube. To compensate for the information loss due to sampling, we select not only the cubes that meet the above criterion but also their neighboring cubes.

## 3.2 NETWORK ARCHITECTURE

As illustrated in Fig. 2, we follow the large reconstruction model to build our transformer network. Our network comprises three components. The first is the Coarse Proposal Network (CPN), which takes images from four viewpoints and learnable cubes as input and predicts coarse cubes around the 3D avatar. The second component involves a sparsification operation, which reduces the number of invalid cubes, thereby significantly enhancing computational efficiency. The last component is the Sparse Cube Transformer (SCT), which decodes the cube features into the corresponding 3D avatar.

**Coarse Proposal Network** We use DINO (Caron et al., 2021) to encode the multi-view images into pose-aware tokens (similar to Li et al. (2023a)). Unlike LRM (Hong et al., 2023) and Instant3D (Li et al., 2023a), we use 3D cubes instead of Triplanes to represent the scene. We first

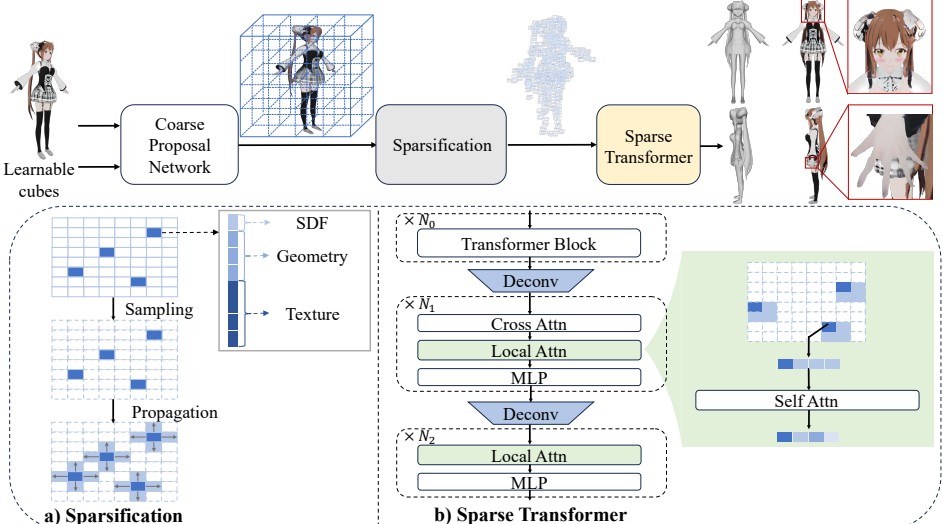

Figure 2: Illustration of the proposed pipeline. Given the learnable cubes and multi-view images, the coarse proposal network predicts the cubes that enclose the avatar. A sparsification operation is applied to reduce the number of invalid cubes. The Sparse Cube Transformer takes these valid cubes as input and decodes the geometric structure and texture of the 3D avatar.

initialize cubes with a learnable 3D shape of $32^3 \times 1024$. Then, the cubes are reshaped into a sequence of learnable tokens. The DINO image tokens are embedded into the 3D cubes through cross-attention, followed by self-attention and MLP layers.

The number of tokens has a significant impact on the generated meshes. High-resolution cubes with more tokens can produce better meshes. In the coarse proposal network, we use cubes of $32^3$ to reduce memory and computational costs. After sixteen 3D transformer blocks, we extract the coarse meshes from the 3D cubes. To improve the extracted mesh resolutions, we use two lightweight deconvolution layers (with a stride of 2) to upsample the cube representation before the mesh extraction. Note that the lightweight upsampling block is used after the transformer and only before the mesh extraction. This makes it memory and computationally efficient.

**Sparsification**    As illustrated in Fig. 2, based on the SDF values of these cubes (Eq. 1), we transform the dense cubes into sparse ones, preserving only valid cubes. To avoid inaccurate meshes and missing parts in the coarse proposal meshes, We also propagate the valid voxels with its neighboring cubes. This propagation is repeated twice to select each neighborhood and its neighborhood's neighborhoods. In this way, a total of $M_k$ cubes are selected, where $M_k \ll M$ and $M$ is the total number of cubes. In our experiments, $M$ is set to $64^3$, and $M_k$ is about 12K, accounting for less than 5% of the total number of cubes.

**Sparse Cube Transformer**    To generate a high-quality 3D avatar from the $M_k$ valid cubes, we design a Sparse Cube Transformer (SCT) (illustrated in Fig. 2). The $M_k$ SparseCubes and the DINO image tokens are fed into a transformer decoder to refine the SparseCubes. The decoder contains three kinds of transformer blocks. After the first and the second transformer blocks, we implement upsampling layers to improve the cube resolution to $256^3$.

For the first part, we implement sixteen transformer blocks similar to Instant3D(Li et al., 2023a). Then, we implement the first upsampling layer and employ eight transformer blocks to optimize the cubes at a higher resolution of $128^3$. In the cross-attention mechanism, the SparseCubes serve as the query sequence, while the image tokens from DINO act as the key-value sequence. Instead of the original self-attention, which is memory and computationally expensive, we implement self-attention in a minimal 3D region (consisting of $2^3$ consecutive cubes). Such a self-attention layer can reduce the memory and computation costs by more than 1000 times. It can also help generate fine details and delicate structures in a local region after upsampling.

Furthermore, we implement another upsampling and refinement block (consisting of one deconvolution layer and four transformer blocks). This refinement block is realized at a resolution of $256^3$. The number of tokens increases significantly, even though we use SparseCubes. Therefore, we remove the cross-attention layer from the transformer block and only implement the self-attention layer in a minimal region of $4^3$. We observe that there is little loss in mesh and texture quality, but our method achieves a 10 times acceleration with 100 times less memory usage.

### 3.3 IMPLEMENTATION DETAILS

**Training Loss** We optimize our model by minimizing the following loss function:

$$
\begin{aligned}
\mathcal{L} = \frac{1}{V} \sum_{i=1}^{V} (&\mathcal{L}_{\text{image}}(\hat{I}_i, I_i^{\text{gt}}) + \lambda_{\text{lpips}} \mathcal{L}_{\text{lpips}} \left( \hat{I}_i, I_i^{\text{gt}} \right) \\
&+ \lambda_{\text{mask}} \mathcal{L}_{\text{mask}}(\hat{M}_i, M_i^{\text{gt}}) + \lambda_{\text{depth}} \mathcal{L}_{\text{depth}}(\hat{D}_i, D_i^{\text{gt}}) + \lambda_{\text{normal}} \mathcal{L}_{\text{normal}}(\hat{N}_i, N_i^{\text{gt}})),
\end{aligned}
\tag{2}
$$

where $V$ is the number of image views, $\hat{I}_i$, $\hat{M}_i$, $\hat{D}_i$ and $\hat{N}_i$ are the rendered image, mask, depth map and normal of each viewpoint, respectively, and $I_i^{\text{gt}}$, $M_i^{\text{gt}}$, $D_i^{\text{gt}}$ and $\hat{N}_i^{\text{gt}}$ are their corresponding ground truths. $\mathcal{L}_{\text{image}}$, $\mathcal{L}_{\text{depth}}$ and $\mathcal{L}_{\text{normal}}$ are $L_2$ loss and $\mathcal{L}_{\text{mask}}$ is an $L_1$ loss. During the training of the Coarse Proposal Network, we set $\lambda_{\text{lpips}}$ and $\lambda_{\text{mask}}$ to 2, and $\lambda_{\text{depth}}$ and $\lambda_{\text{normal}}$ to 1. For the training of the Sparse Cube Transformer, we set $\lambda_{\text{lpips}}$ and $\lambda_{\text{normal}}$ to 1, $\lambda_{\text{mask}}$ to 8, and $\lambda_{\text{depth}}$ to 20.

**Learning Strategies and Computational Resources** The whole network can be trained in an end-to-end manner. However, for faster training, we first optimize the coarse proposal network using Eq. 2 for 121 hours with 32 A100 GPUs (30k iterations). The batch size is 5 and the learning rate is 4e-4 with a cosine decay. After that, we warm up the Sparse Cube Transformer with a learning rate of 4e-4. We also use the upsampled output of the coarse proposal network as the learning target and train the Sparse Cube Transformer with $L_2$ loss for 14k iterations. Finally, we start to optimize the Sparse Cube Transformer using the same loss as the coarse proposal network with a smaller learning rate of 5e-5 and a batch size of 2. Learnable positional embeddings are used in both stages. The shapes of the positional embeddings in the first and second stage are $32^3 \times 1024$ and $64^3 \times 1024$, respectively. During the training phase, the positional embeddings for the second stage are initialized by upsampling the positional embeddings used in the first stage. After getting the relative spatial positions of valid SparseCubes, we can select valid tokens from the learnable positional embeddings as the input of the second stage. LRM (Hong et al., 2023) requires 9,000 A100 GPU hours for training, while our algorithm requires a total of 3,800 A100 GPU hours, less than half of LRM's requirement.

**Datasets and Augmentations** We use 20K anime characters to train our models. Limited by the available amount of 3D data, we find it difficult to generate high-quality representations for some uncommon characters (e.g., characters with wings). We observe that most of our character data is centered within the cube, with only a small portion deviating from the center due to large tails or wings. For this reason, position embeddings that are not centered within the cube did not receive effective optimization, resulting in poor generation outcomes for characters with large tails or wings. Therefore, we designed a data augmentation method to randomly displace character data in the front, back, left, and right directions. This ensures that position embeddings not centered within the cube also receive effective optimization, leading to significantly improved results.

**Multi-view Back Projection** To further improve the texture accuracy of our generated mesh, we employed a texture back-projection strategy similar to that in Peng et al. (2024). Since our generated mesh is highly accurate and best matches the input multi-view images, there are very few conflicting areas during the back-projection process.

## 4 EXPERIMENTS

In this section, we conduct both qualitative and quantitative experiments to compare our method with existing state-of-the-art 3D generation methods.

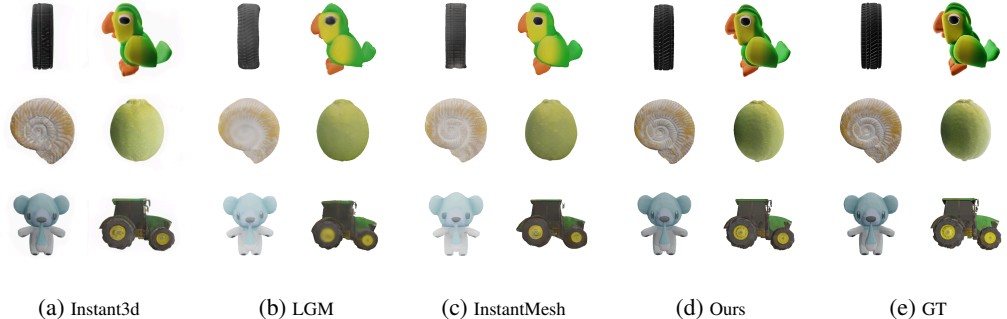

| (a) Instant3d | (b) LGM | (c) InstantMesh | (d) Ours | (e) GT |
|---|---|---|---|---|

Figure 3: Qualitative Comparisons on the LVIS subset of the Objaverse dataset (Deitke et al., 2023). Each of the columns shows the results of a method (from left to right: Instant3D (Li et al., 2023a), LGM (Tang et al., 2024a), InstantMesh (Xu et al., 2024), Ours and ground truth). Our method is able to generate significantly more detail for general 3D objects.

Table 1: Comparisons on ground truth single image to 3D. To ensure the fairness of the evaluation, we fine-tune all the methods used for evaluation on our dataset of 20K anime characters. The test set includes data of 30 randomly selected anime characters. We calculate PSNR and SSIM on eight full-body images and four headshot images from different perspectives. * For DreamGaussian and InstantMesh, we use the provided pre-trained model without fine-tuning.

| Method | PSNR (%)↑ | SSIM↑ | CD ($10^{-5}$) (%)↓ | Time (s) ↓ |
|---|---|---|---|---|
| DreamGaussian (Tang et al., 2024b)* | 15.87 | 0.8263 | 184.96 | 180 |
| InstantMesh (Xu et al., 2024)* | 14.76 | 0.8460 | 492.48 | **9** |
| Instant3D (Li et al., 2023a) | 23.19 | 0.9106 | 19.42 | 15 |
| Instant3D with DMTet (Peng et al., 2024) | 20.40 | 0.8915 | 48.26 | 10 |
| LGM (Tang et al., 2024a) | 24.05 | 0.9213 | 28.49 | 52 |
| Ours | **24.99** | **0.9283** | **5.90** | 20 |

## 4.1 COMPARISONS WITH STATE-OF-THE-ARTS

We compare our method against recent techniques capable of generating 3D meshes from a single image. The comparison results are shown in Fig. 4 and Fig. 5. All input images are produced using the fine-tuned PIXART-$\Sigma$ (Chen et al., 2024). For each generated mesh, we visualize both the entire body (upper) and specific body parts (lower). The face and hand regions are rendered using Blender (Community, 2018) to allow for detailed visual comparisons. Our method produces 3D meshes with superior visual effects and high-quality details. The meshes we generate exhibit vivid and lifelike facial features, while those generated by other methods often appear blurry. Additionally, our meshes feature distinct and well-separated fingers, whereas meshes generated by other methods frequently show merged hands.

We train Instant3D and LGM on our training set under the same settings and then compare our method with them. We also compare our method with DreamGaussian (Tang et al., 2024b) and InstantMesh (Xu et al., 2024) by using the provided pre-trained models.

**Comparisons on the anime avatar dataset** Tab. 1 presents the comparison results. To ensure a fair comparison we do not use the texture back-projection strategy for any of the algorithms when calculating these metrics. Our algorithm achieved the best PSNR and SSIM, indicating that the generated results of our method have the highest visual quality. In addition, our method significantly outperforms other approaches in terms of Chamfer Distance (CD) loss. Compared with the second-best method, Instant3D, our method reduces Chamfer Distance error by about 70%, indicating that our approach produces much higher mesh quality and better details. As shown in Tab. 1, our algorithm is also very fast in terms of generation speed. We also compare our method with an SMPL-based approach, TeCH (Huang et al., 2024). The result is shown on Fig. 9 of the Appendix.

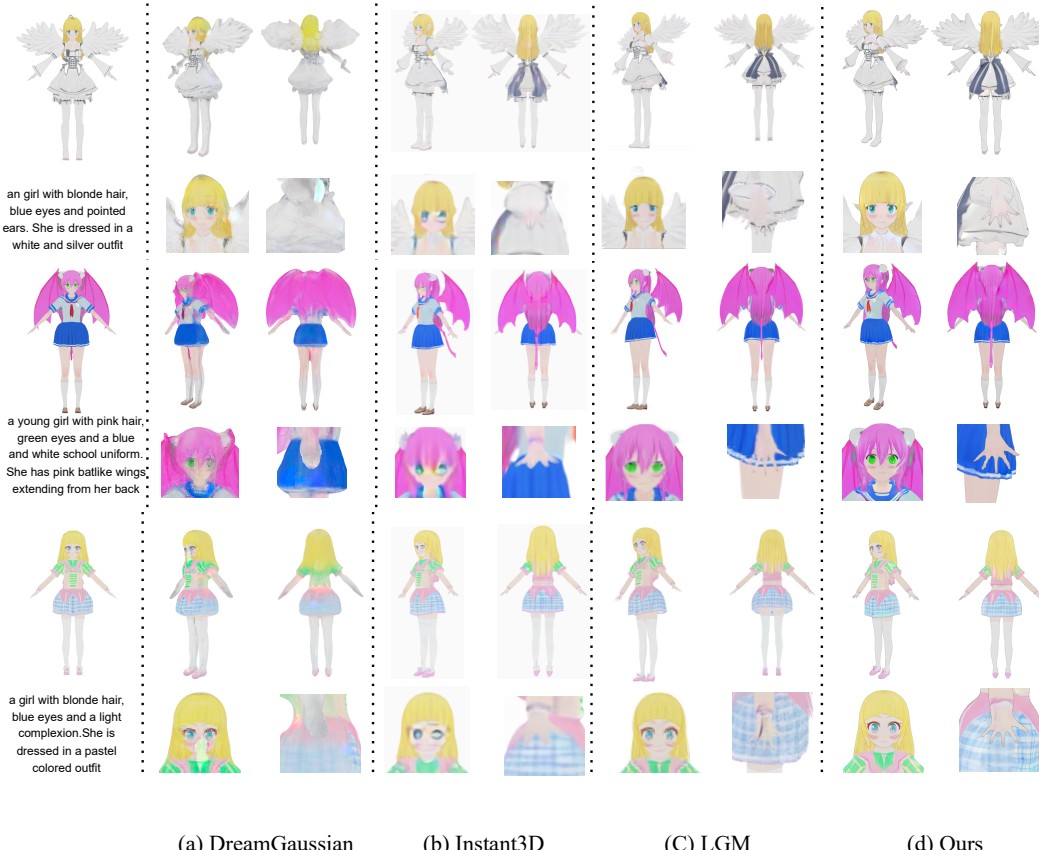

|              (a) DreamGaussian           (b) Instant3D           (C) LGM           (d) Ours              |

Figure 4: Qualitative Comparisons. *First column*: We use the text prompts to generate the input image. Each of the remaining columns shows the results of a method (from left to right: Dream-Gaussian (Tang et al., 2024b), Instant3D (Li et al., 2023a), LGM (Tang et al., 2024a), and Ours). To compare the quality of details and delicate structures, we also show zoomed-in views of the hand and face regions.

Table 2: Comparisons with other methods on the LVIS subset of the Objaverse dataset (Deitke et al., 2023). The test set includes data from 30 randomly selected objects.

| Method | PSNR (%)↑ | SSIM↑ | CD ($10^{-5}$) (%)↓ |
|---|---|---|---|
| InstantMesh (Xu et al., 2024) | 23.67 | 0.9129 | 71.32 |
| Instant3D (Li et al., 2023a) | 23.16 | 0.9097 | 81.09 |
| Instant3D with DMTet (Peng et al., 2024) | 22.00 | 0.9063 | 230.64 |
| LGM (Tang et al., 2024a) | 24.70 | 0.9389 | 116.14 |
| Ours | **26.78** | **0.9452** | **36.86** |

**Comparisons on the realistic human dataset**  We test our method on the RenderPeople dataset (ren, 2018) to demonstrate that it can generate high-quality, realistic-style characters. The experimental results in the Tab. 5 of the Appendix demonstrate the effects of our method. Our Sparse Transformers achieve the best PSNR for evaluating the rendering qualities and the best CD loss for evaluating the mesh qualities.

**Comparisons on the LVIS subset of the Objaverse dataset**  Since most algorithms (Tang et al., 2024b; Xu et al., 2024; Li et al., 2023a; Tang et al., 2024a), are designed for the general 3D object generation task, in order to ensure a fair comparison with these methods, we train our method on the LVIS subset of the Objaverse dataset(Deitke et al., 2023). Tab. 2 shows the quantitative results. Our method outperforms other methods across all metrics, demonstrating its ability to generate high-

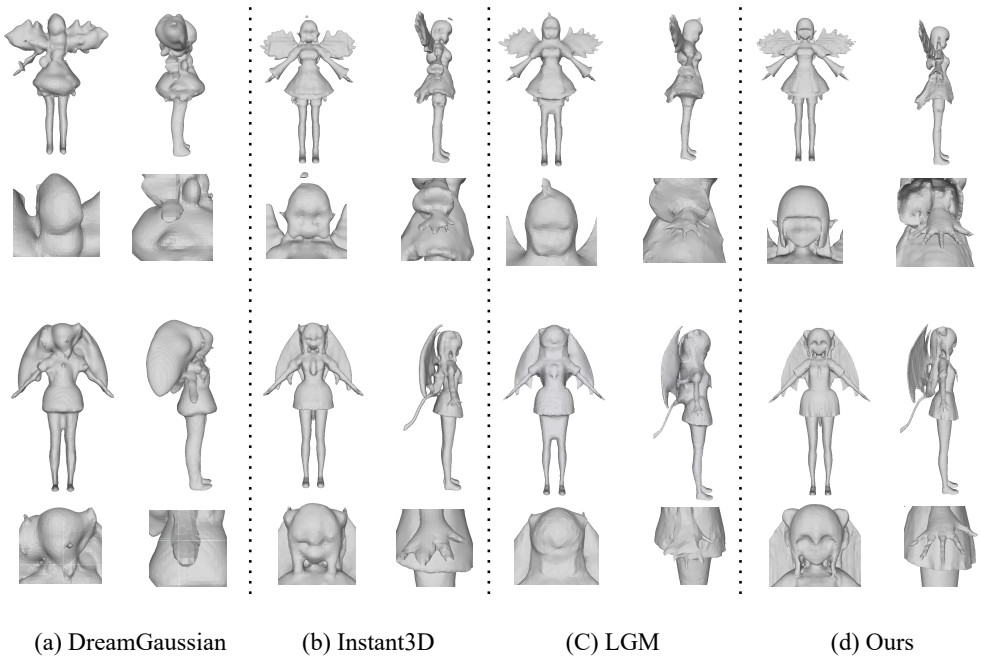

(a) DreamGaussian      (b) Instant3D      (C) LGM      (d) Ours

Figure 5: Mesh Qualitative Comparisons. Characters are from Fig. 4. Additional results are provided in the Appendix. Our method far outperforms state-of-the-art methods in mesh quality.

Table 3: Comparisons between SparseCubes and Dense Cubes representations. We report the speed and memory usage during the training phase with a batch size of 1.

| Type | Resolution | PSNR↑ | SSIM↑ | CD($10^{-5}$)↓ | MEM(GB) | Time per iter (s) | Number of faces (K) |
|---|---|---|---|---|---|---|---|
| Dense | 64 | 21.34 | 0.8987 | 19.88 | 16 | 40 | 20 |
| Dense | 128 | - | - | - | 24 | 340 | 50 |
| Dense | 256 | - | - | - | 80 | 2850 | 170 |
| Sparse | 64 | 21.08 | 0.8970 | 13.87 | 8 | 1 | 20 |
| Sparse | 256 | **24.99** | **0.9283** | **5.90** | 37 | 5 | 170 |

quality 3D objects. The generated results of our method have the highest visual quality, as shown in Fig. 3 and Fig. 8.

**SparseCubes versus Implicit Representation** Instant3D (Li et al., 2023a) and LGM (Tang et al., 2024a) encode the 3D space into an implicit NeRF (TriPlanes) or 3D Gaussian and utilize transformers to learn the meshes and textures. As shown in Tab. 1, Fig. 4, and Fig. 12, methods with implicit representations use strong smoothness to remove floating artifacts; as a result, fine details and delicate structures are missing in the predicted results. Our SparseCubes use 3D representation and 3D transformers in learning, which can better preserve geometric features and generate high-quality meshes with exceptional detail and intricate structures, particularly in crucial areas such as faces and hands.

**SparseCubes versus DMTet/FlexiCubes** DMTet and FlexiCubes use dense cubes as 3D mesh representations. Instant3D with DMTet and InstantMesh with FlexiCubes use dense 3D cubes that require much more memory and computational costs than our SparseCubes. As shown in Tab. 1, they can only run with a low-resolution ($128^3$) cube representation. The generated results do not have a large Chamfer Distance error. Additionally, we observe that InstantMesh and Instant3D with DMTet generate many holes in light-colored regions (as shown in Fig. 7 of the Appendix).

Table 4: Analysis of data augmentation. For faster training and evaluation, we use the model with a resolution of 64.

| Random displacement | PSNR $\uparrow$ | SSIM $\uparrow$ | CD $(10^{-5}) \downarrow$ |
|---|---|---|---|
| | 19.81 | 0.8856 | 53.8 |
| $\checkmark$ | **22.11** | **0.8997** | **46.1** |

**SparseCubes verse Voxels**  Voxel-based methods that discretize 3D space as $0$ and $1$ (Ren et al., 2024; Schwarz et al., 2022) tend to generate meshes with voxel effects (as illustrated in Fig. 6). Our SparseCubes can produce smooth mesh surfaces with intricate structures.

## 4.2 ABLATION STUDY

**SparseCubes versus Dense Cubes**  Tab. 3 presents the comparison results between using SparseCubes and Dense Cubes representations. We use the same number of transformer blocks and compute the memory usage and speed with a batch size of one.

At the same resolution of 64, the PSNR and SSIM of the sparse model are comparable to those of the dense model, indicating that the degradation in visual quality introduced by our sparsification operation is negligible. Meanwhile, the Chamfer Distance of the sparse model is much better than that of the dense model, demonstrating that the proposed sparsification enables the model to focus more on the valid positions, thus resulting in more accurate geometry. In addition, the sparse model run 40 times faster than the dense one. When increasing the resolution to 256, our sparse model achieves far better accuracy in both textures and meshes.

Employing Dense Cubes with a resolution of 256 significantly increases GPU memory consumption (by 2 times) and training time (by more than 100 times), thereby hindering the acquisition of timely training outcomes. Training with a resolution of 128 is also difficult as it consumes 64 times more computational resources than our Sparse Cube Transformer.

Note that, the FlashAttention (Dao et al., 2022) in the xFormers (Lefaudeux et al., 2022) reduce the memory costs in training transformers at the expense of increased runtime. The actual memory costs of the dense models (for both resolutions of 128 and 64) are much higher than those of our Sparse Cube transformers.

**Random Displacement**  We conduct further analysis to assess the impact of random displacement on the performance of our algorithm. Experiments are performed using a model with a resolution of 64. The results in Tab. 4 indicate that our data augmentation method significantly improves the quality of the generated results.

## 5 CONCLUSION

In this work, we introduced SparseCubes, a novel and efficient sparse differentiable mesh representation method, along with a sparse transformer network for generating high-quality 3D characters. Our approach addresses the limitations of current state-of-the-art 3D generation methods, which struggle to capture fine details and delicate structures crucial for applications such as anime avatar generation. Through extensive testing and validation, we demonstrated that SparseCubes significantly reduce computational requirements and achieves high-quality and highly detailed 3D mesh generation.

**Limitations and Future work**  Limited by the amount of available 3D data and the diversity of text descriptions, the fine-tuned text-to-image model currently doesn't always perfectly match the textual descriptions. We believe this can be improved by fine-tuning more powerful diffusion transformer models (Li et al., 2024). Additionally, providing more diverse and longer text prompts can also boost the alignment. As for future work, we will explore ways to further increase the resolution of the SparseCubes (*e.g.,* $1k^3$ with millions of mesh faces) to achieve hyperfine structures in 3D meshes.

## ETHICS STATEMENT

Our work shares similar concerns with other generative modeling efforts, particularly regarding the potential for misuse in creating misleading content. Additionally, biases present in the training datasets can be reflected in the generated outputs. As with any powerful creative tool, there is a risk of misuse. Generated characters could be used inappropriately or to create harmful content. It is essential to develop and enforce ethical guidelines and content moderation strategies to mitigate such risks.

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

## A  APPENDIX / SUPPLEMENTAL MATERIAL

**Ablation study on other parameters**  Since training large-scale transformer models costs signif-icant GPU resources, ablation on all kinds of parameter settings (*e.g.,*the balance weights of the loss function and the learning rate) is impossible. For the remaining parameters, we adjust them only in a fine-tuning manner. Specifically, we fine-tune the model with different parameter settings (*e.g.,*balance weights) for just 1000 iterations (about 6 hours) to discover the best settings by observ-ing its convergence speed and CD loss of the meshes.

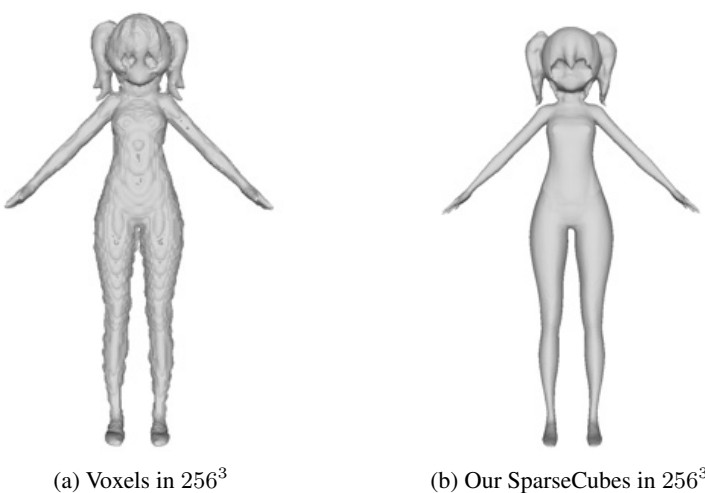

(a) Voxels in $256^3$      (b) Our SparseCubes in $256^3$

Figure 6: Comparison of mesh results between our SparseCubes (at a resolution of $256^3$) and voxels. (a) There are clear voxel effects in the mesh produced by voxelization. (b) Our SparseCubes can produce high-quality meshes at the same resolution.

**More results**

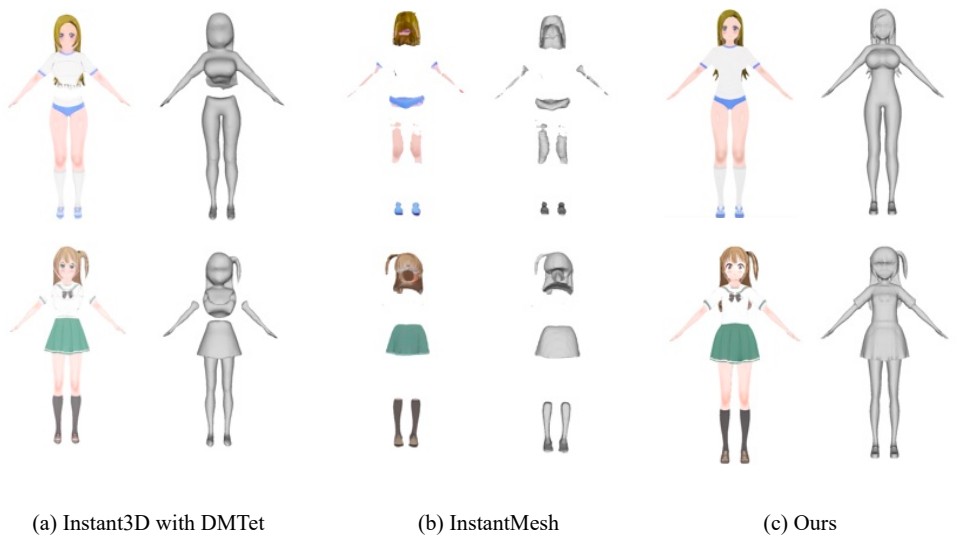

(a) Instant3D with DMTet       (b) InstantMesh       (c) Ours

Figure 7: Illustration of holes in light-colored regions using Instant3D with DMTet and InstantMesh (without fine-tuning)

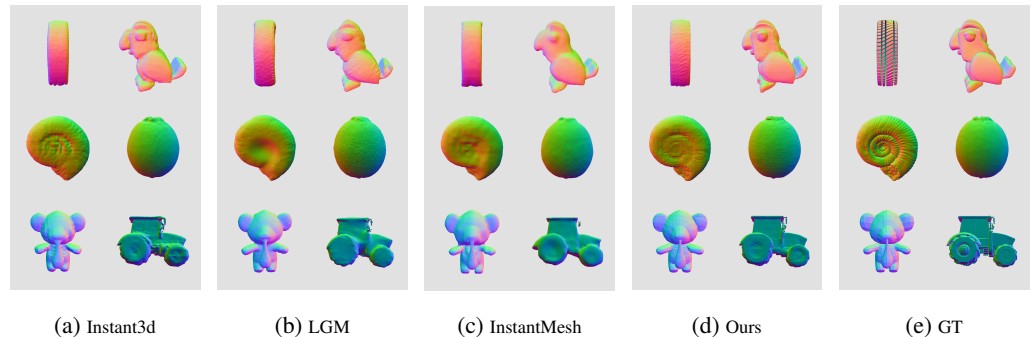

(a) Instant3d     (b) LGM     (c) InstantMesh     (d) Ours     (e) GT

Figure 8: Results of normal on the LVIS subset of the Objaverse dataset (Deitke et al., 2023). Each of the columns shows the results of a method (from left to right: Instant3D (Li et al., 2023a), LGM (Tang et al., 2024a), InstantMesh (Xu et al., 2024), Ours and ground truth). Our method is able to generate significantly more detail for general 3D objects.

Table 5: Comparison on RenderPeople ren (2018) dataset.

| Method | PSNR (%)↑ | CD ($10^{-5}$) (%)↓ |
|---|---|---|
| Instant3D Li et al. (2023a) | 18.79 | 15.00 |
| Instant3D with DMTet Peng et al. (2024) | 18.32 | 34.81 |
| LGM Tang et al. (2024a) | 20.02 | 11.78 |
| Ours | **25.93** | **9.60** |

Tech

ImageDream

Ours

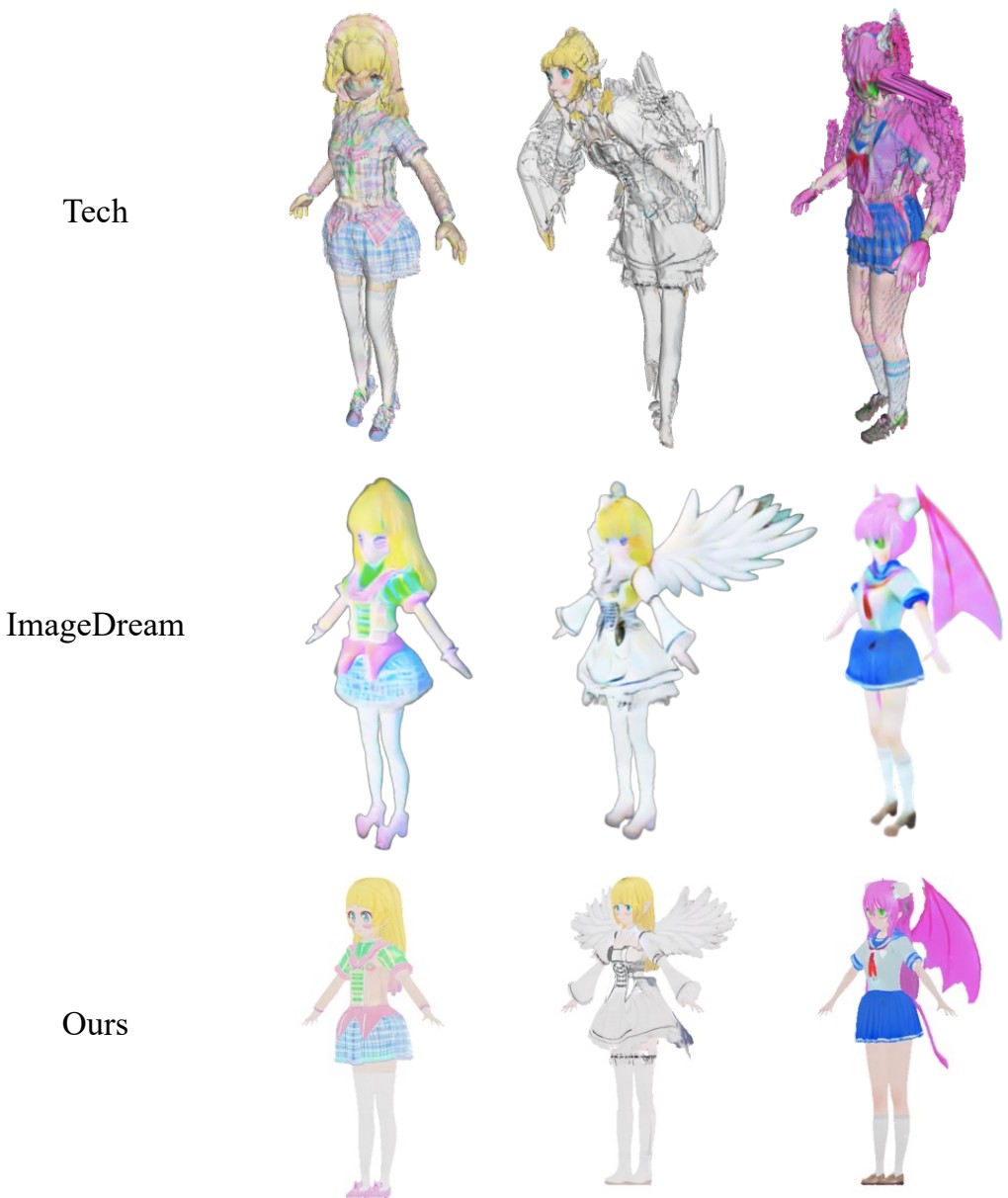

Figure 9: Compairsons with the shapes of Tech (Huang et al., 2024) and ImageDream (Wang & Shi, 2023). The TeCH method (Huang et al., 2024) heavily depends on the SMPL model. However, the 3D character data includes diverse hairstyles and outfits that the SMPL model cannot adequately represent. Consequently, the characters generated by TeCH exhibit significantly poor quality.

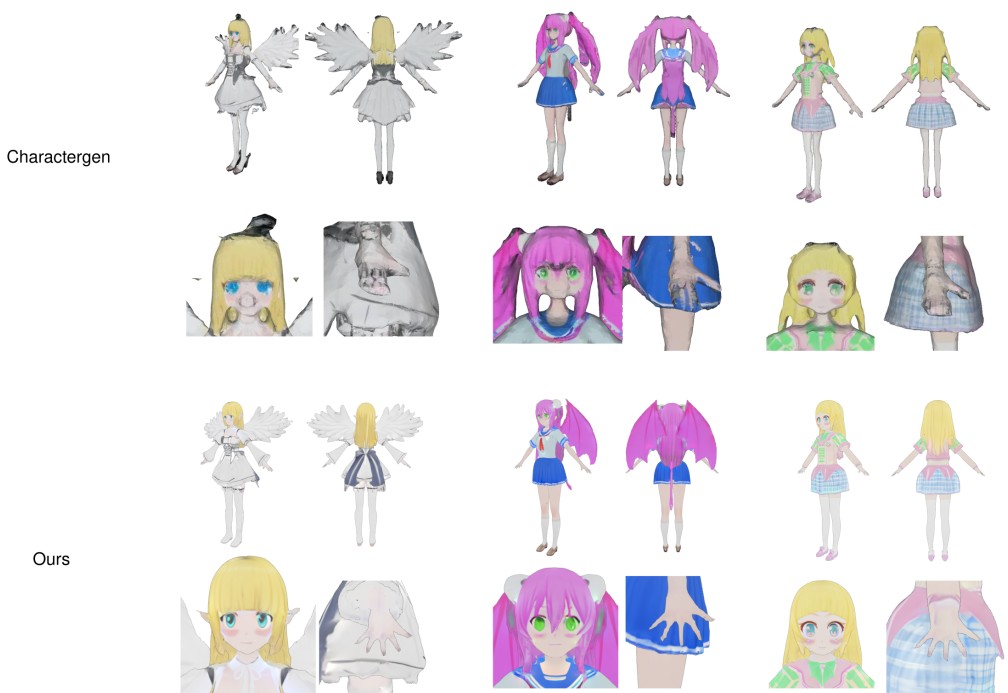

Figure 10: Comparisons with CharacterGen (Peng et al., 2024). We have conducted a comparison between CharacterGen and our method, as shown in Tab. 1. Specifically, we implemented Instant3D with DMTet, the approach adopted by CharacterGen. Here we utilize the weights released by CharacterGen for further evaluation. The results clearly demonstrate that our method significantly outperforms CharacterGen, producing meshes with finer details and textures that are noticeably sharper and more realistic.

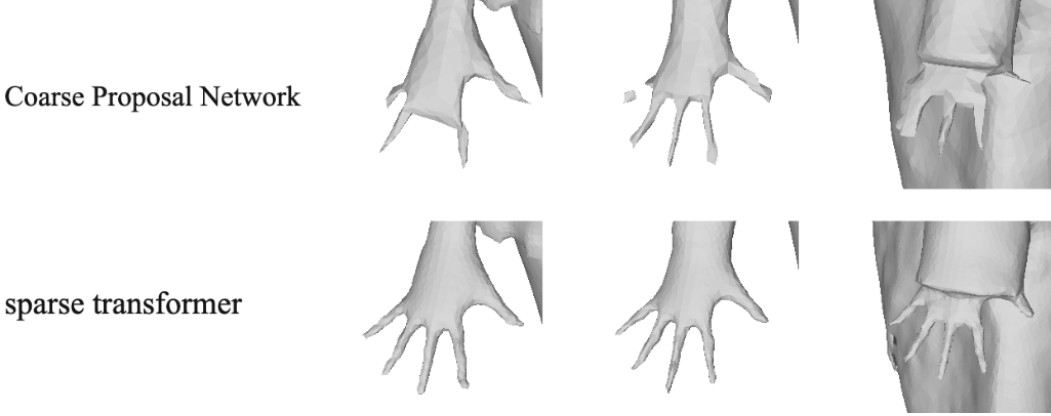

Figure 11: Comparisons between the coarse proposal network and the sparse transformer. We compared the quality of the finger meshes generated by the coarse proposal network and the sparse transformer. It is evident that the sparse transformer significantly improves the quality of the fingers in the generated meshes.

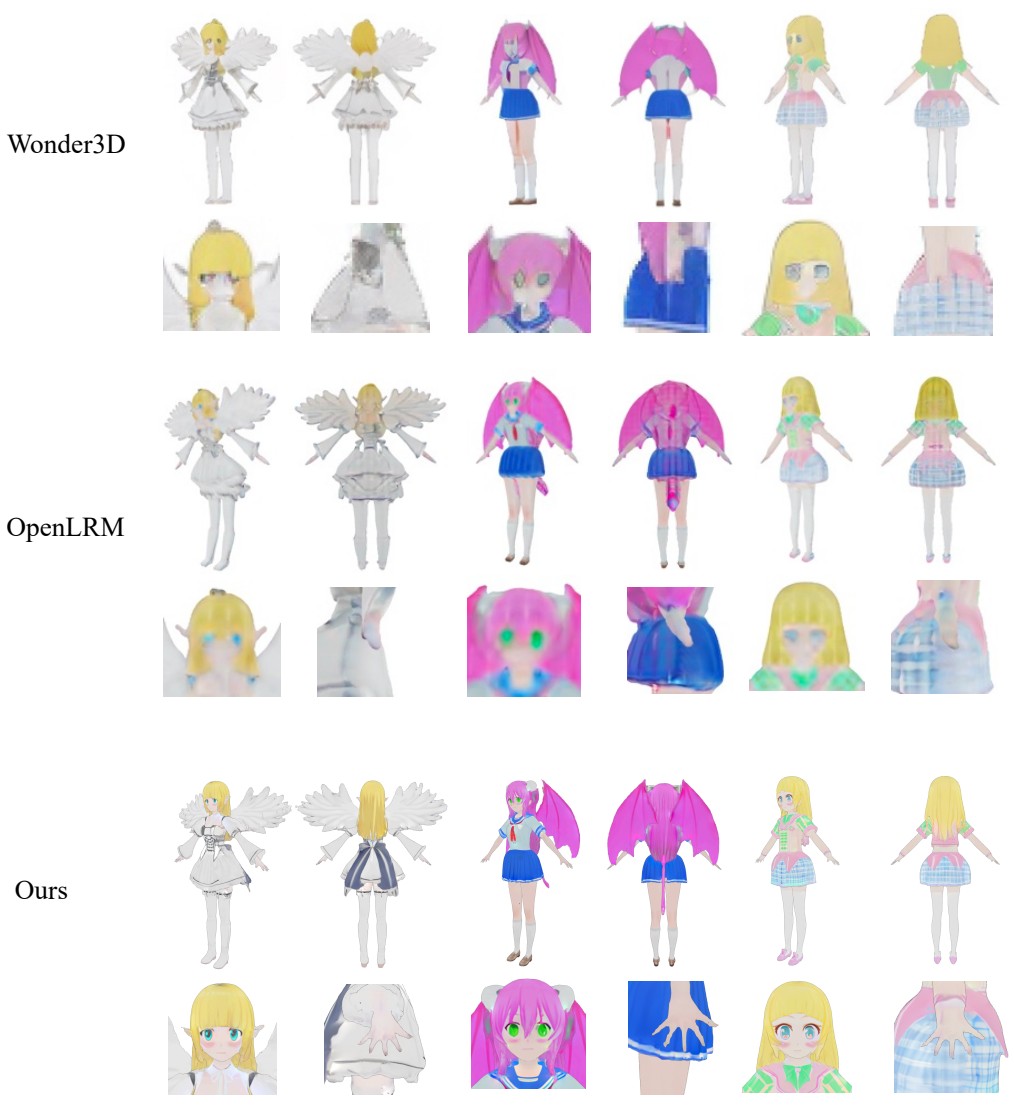

Figure 12: Visual Qualitative Comparisons. Extra comparisons of Fig. 4

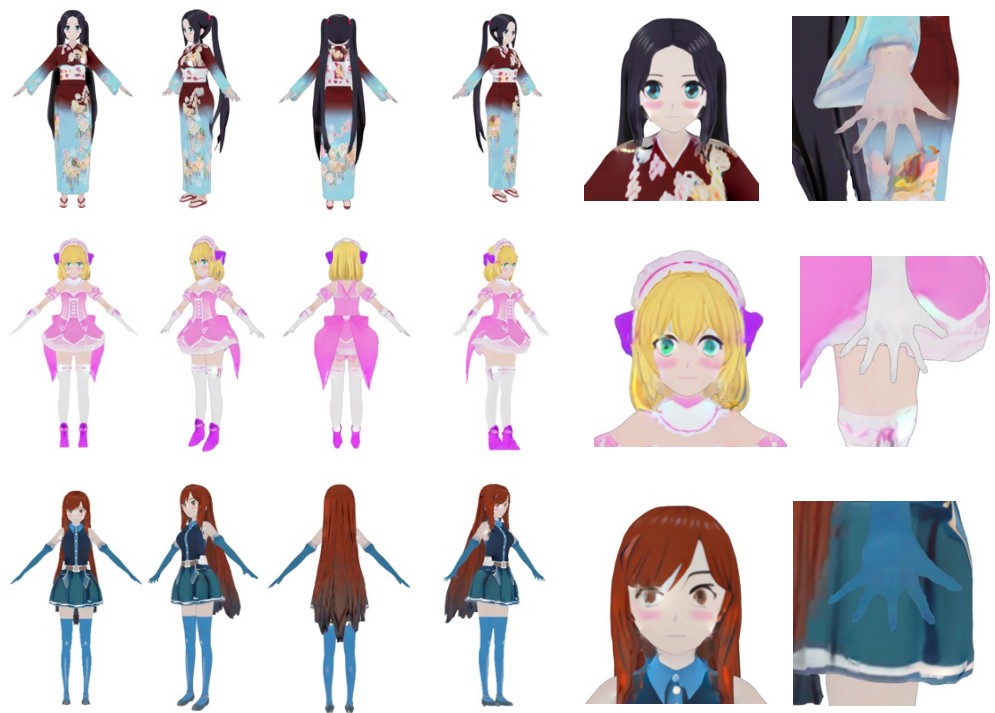

Figure 13: More results

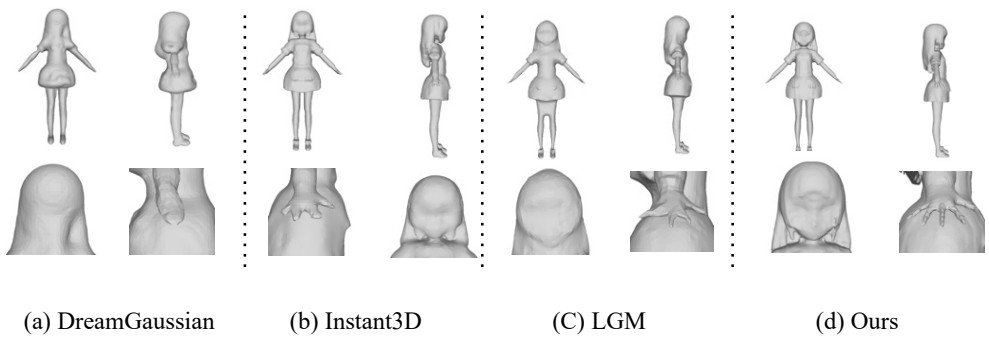

(a) DreamGaussian      (b) Instant3D      (C) LGM      (d) Ours

Figure 14: Additional mesh results from Fig. 5

