# OpenReview forum: "High-quality Text-to-3D Character Generation with SparseCubes and Sparse Transformers."
_ICLR.cc/2025/Conference — ICLR 2025 Poster_

### Official Review · Reviewer_ozD8 · 2024-11-01

**Soundness:** 2
**Presentation:** 3
**Contribution:** 1
**Rating:** 5
**Confidence:** 3

**Summary:**

This paper targets to solve the high-resolution mesh generation problem with text prompts.
It argues the current text-to-mesh methods are mainly bottlenecked by the cubical memory usage to generate high-resolution meshes,
which leads to the limitation of the mesh resolution, and the quality of the generated meshes.
This problem is believed to be in particular severe for human-like/character-based meshes, like fingers, hair, fur, etc.

To solve this problem, this paper introduce the sparseCube and the sparse transformer.
It first generates a low-resolution (32x32x32) grid to obtain a mesh in a differentiable manner, and then sparsifies the grid to get a sparse grid.
Finally, it uses a sparse transformer to generate the high-resolution mesh from the sparse grid.
This paper compares the proposed method on objaverse and human character datasets, and demonstrating the effectiveness of the proposed method.

**Strengths:**

1. The overall pipeline is sound and reasonable. Utilizing the sparsity of voxel grid, we can pursue the high-resolution meshes (using FlexiCubes), leading to high-resolution 3D assets generation.
2. This paper is well written: the story and method is easy to follow and understand.
3. Results look great.

**Weaknesses:**

Overall, the contribution of this project is OK but mild.
This paper utilizes the idea of Flexicubes to train a model to predict the 3D voxels for a mesh, and renders the 2D images from the 3D mesh in a differentiable manner, which is a still in the scope of Flexicubes.
The main contribution of this paper should be utilizing a sparse transformer to do refinement or upsampling for the 3D voxels after pruning.
This is a sound idea while not novel.

Some weakness:

1. I think there are some baselines missing. In particularly, XCUBE [XCube: Large-Scale 3D Generative Modeling using Sparse Voxel Hierarchies]. Although XCUBE uses diffusion model for generation and its output voxel grids are supervised by 3D loss, but it also highlights a sparse voxel grid solution (a novel data structure and companying network to process it). I am curious can you use XCUBE's sparse voxel grid method (fvdb) to do the same thing as yours? And compare the speed, memory and performance difference? I am very curious about how the proposed method. is compared again the XCUBE's backbone. Also, SparseConv-Unet might be another reasonable  backbone to compare against the Sparse Transformer.

2. In table 1, the proposed method takes 20 s to generate, which is slower than InstantMesh. I am curious as a feed-forward model, why it is slower? And what's the time using ratio of each component?

**Questions:**

1. I am a little bit confused about the relationship between 3D character generation and sparse CUBE/Transformer. For sparse cube/transformer, it seems they can work on any 3D content generation without any specific constraints like character. So I am curious why this paper mainly argues about 3D character generation? I can understand 3D character has some very thin structures like hair, fingers, but I still think generating high-quality 3D mesh would be a common interest for 3D generation, whatever the objective is.
2. Just for clarification, the renderer (from 3D mesh to image) is the same as FlexiCubes or not?
3. For sparse transformer, do you apply position encoding (more specifically, the 3D location/index of occupied cubes) ? If so, how do you do that?
4. It would be helpful to evaluate the results just after the coarse proposal network, and after prunning, to show how much gains we get from the sparse Transformer as a refinement stage.

---

> ### Author Response · Authors · 2024-11-20
> **Rebuttal To Reviewer ozD8**
>
> Thank you for your valuable suggestions and comments! We have tried to address your concerns in the rebuttal text:
>
> **Contribution.**
>
> Previous methods like Instant3D [1] and InstantMesh [2] relied on triplane representations to train large reconstruction models. However, the triplane approach limits the ability to implement sparse methods. Directly using high-resolution voxels is extremely challenging due to computational resource and memory overhead, as shown in Table 3 of our paper. Our method effectively addresses this issue by significantly reducing both computational cost and memory usage.
>
> **Differences Between Our Method and XCUBE.**
>
> Although both XCUBE and our method utilize sparse voxels, there are significant differences. As you mentioned, XCUBE is supervised by 3D losses, meaning that the sparse voxel hierarchies are pre-defined and can be directly used for training. In contrast, our method is only supervised by 2D losses, with ground truth limited to 2D images and no known 3D structure. During training, we derive sparse information solely from 2D images, whereas XCUBE requires 3D priors to obtain sparse information. This is the essential distinction between the two methods.
>
> **Relationship Between 3D Character Generation and Sparse Cubes/Transformers.**
>
> We have demonstrated that our method also achieves state-of-the-art performance on a general object dataset (LVIS subset). However, the primary focus of our method is 3D character generation, as characters often feature very thin structures like hair and fingers, as you noted. These structures provide a challenging testbed to showcase the effectiveness of our approach.
>
> **Implementation Details.**
>
> The rendering process is the same as in FlexiCubes.
> Learnable positional embeddings are employed in both stages. The shapes of the positional embeddings for the first and second stages are $32^3 \times 1024$ and $64^3 \times 1024$, respectively. During training, the second-stage positional embeddings are initialized by upsampling those used in the first stage. In the cross-attention mechanism, the positional embeddings act as the query sequence, while image tokens from DINO serve as the key-value sequence.
>
> **Inference Time**
>
> Since the final sparse cube resolution we achieve is 256, the resulting mesh contains significantly more faces than InstantMesh [2]. Consequently, mesh extraction and UV unwrapping require more time, making our method slower than InstantMesh. Specifically, inference for the coarse proposal network and the sparse transformer together takes approximately 5 seconds, mesh extraction requires 5 seconds, and UV unwrapping takes 10 seconds.
>
> **Comparison Between the Coarse Proposal Network and the Sparse Transformer.**
>
> We evaluated our method using the same test set as in the experiments of Table 1 in our paper, with the results  shown in Table 1 below. Additionally, we compared the quality of the finger meshes generated by the coarse proposal network and the sparse transformer. The results, presented in Figure 11 of the appendix n our paper, clearly show that the sparse transformer significantly improves the quality of the fingers in the generated meshes.
> | Stage                                  | PSNR$\uparrow$      | SSIM$\uparrow$       | CD ($10^{-5}$)$\downarrow$ |
> | -------------------------------------- | ------------------- | -------------------- | -------------------------- |
> | Coarse Proposal Network                | 22.39               | 0.9124               | 6.68                    |
> | Sparse Transformer                       | 24.99              | 0.9283              | 5.90                     |
>
> Table 1: Comparison between the coarse proposal network and the sparse transformer.
>
> [1] Jiahao Li, Hao Tan, Kai Zhang, Zexiang Xu, Fujun Luan, Yinghao Xu, Yicong Hong, Kalyan Sunkavalli, Greg Shakhnarovich, and Sai Bi. Instant3d: Fast text-to-3d with sparse-view generation and large reconstruction model. arXiv preprint, 2023.
> [2] Jiale Xu, Weihao Cheng, Yiming Gao, Xintao Wang, Shenghua Gao, and Ying Shan. Instantmesh: Efficient 3d mesh generation from a single image with sparse-view large reconstruction models. arXiv preprint, 2024.

---

> ### Comment · Reviewer_ozD8 · 2024-11-23
> **Reply to Rebuttal**
>
> Thanks for the rebuttal and explanations.
>
> There might be some misunderstanding in this question, Differences Between Our Method and XCUBE. I totally understand the difference between your settings and XCUBE. My point is: I am curious how the proposed methods compared to other sparse-based representations. You can do the exactly the same thing as you do while using XCUBE's backbone. Would it be feasible? And will there be advantages? There may not be enough time to do it now.

---

> > ### Author Response · Authors · 2024-11-25
> > **Reply to Reviewer ozD8**
> >
> > Thank you for your explanation of this question, Differences Between Our Method and XCUBE.
> >
> > We believe that applying the xcube method to our approach is feasible. Specifically, we can train a decoder to subdivide and prune the cubes generated by the Coarse Proposal Network, and then proceed to the second stage of training. The advantage of this approach is that it can remove some redundant cubes, leading to a slight reduction in the computational cost during the second stage. However, we believe that the number of cubes eliminated is minimal. We believe that the slight increase in computational cost from skipping pruning is more cost-effective than training an additional decoder to handle subdivision and pruning. Furthermore, adding such a decoder would not only extend inference time but also introduce additional errors during the inference process.

---

### Official Review · Reviewer_KMF3 · 2024-11-03

**Soundness:** 2
**Presentation:** 2
**Contribution:** 3
**Rating:** 6
**Confidence:** 4

**Summary:**

This paper introduces SparseCubes, a new method for improving text-to-3D generation of anime characters. SparseCubes uses a sparse mesh representation that cuts computational needs by over 95% and storage by 50%, allowing for better detail in 3D models. The authors demonstrate its effectiveness in accurately rendering delicate features in anime character generation.

**Strengths:**

1. This paper presents the SparseCubes representation, which can capture fine details like fingers and hair.

2. A corresponding transformer network is designed for the SparseCubes representation

3. A significant efficiency improvement compared to previous methods

**Weaknesses:**

Method:

1. High-resolution voxels have already been used to generate smooth meshes in DMTet and FlexibleCube. Improving efficiency and increasing resolution through a sparse design is a straightforward idea, which somewhat affects the novelty of the work.

Experiments:

1. The paper lacks a comparison of results using FlexibleCube as a post-processing step.

2. The experimental results are somewhat confusing: First, the quantitative improvements in PSNR and SSIM on the anime avatar dataset are quite small, and an explanation for this is needed. Additionally, the qualitative comparisons provided for the LVIS subset make it difficult to discern clear differences.

3. The paper lacks a comparison with the current state-of-the-art method, CharacterGen[1], including both qualitative and quantitative comparisons.

4.  For the qualitative results of the anime avatar dataset, the main text focuses on only a few selected IDs. It would be beneficial to include comparisons for more IDs in the experiments.

[1] Charactergen: Efficient 3d character generation from single images with multi-view pose canonicalization.

**Questions:**

1. Is the entire training process conducted in stages, or is it an end-to-end process?

2. The results on the real human dataset only include quantitative metrics and lack qualitative results for comparison.

---

> ### Author Response · Authors · 2024-11-20
> **Rebuttal To Reviewer KMF3**
>
> Thank you for your valuable suggestions and comments! We have tried to address your concerns in the rebuttal text :
>
> **Novelty of Our Work**
>
> Previous works like CharacterGen [1] and InstantMesh [2], did not utilize high-resolution voxels. Instead, they relied on triplane representations. Directly using high-resolution voxels is extremely challenging due to the computational and memory overhead, as shown in Table 3 of our paper. Our method effectively addresses this issue, significantly reducing both computational cost and memory requirements.
>
> **Results Using FlexiCubes as a Post-Processing Step.**
>
> The results presented in our paper are the outputs obtained using FlexiCubes.
>
> **Quantitative Improvements in PSNR and SSIM.**
>
> Using of mask loss, depth loss, and normal loss focuses on generating higher-quality meshes, but slightly reduce PSNR and SSIM. To address this, we employed a multi-view back-projection approach to further enhance texture quality, as discussed in the last subsection of Section 3.3.
>
> **Comparison with CharacterGen.**
>
> We conducted a detailed comparison between CharacterGen and our method, as shown in Table 1 of our paper. Specifically, we implemented Instant3D with DMTet, the approach adopted by CharacterGen [1]. For further evaluation, we also utilized the weights released by CharacterGen. The results, presented in Figure 10 in the appendix, clearly demonstrate that our method significantly outperforms CharacterGen, producing meshes with finer details and textures that are noticeably sharper and more realistic.
>
> **Training Strategies.**
>
> The entire network can be trained in an end-to-end manner. However, to expedite training, we first optimize the coarse proposal network, then use it to warm up the second stage, and finally optimize the entire network, as described in the second subsection of Section 3.3.
>
> **More Qualitative Results.**
>
> Due to the file size limitations of the PDF, we were unable to include more qualitative results. However, additional results can be viewed in the videos provided in our supplementary material.
>
>
> [1] Hao-Yang Peng, Jia-Peng Zhang, Meng-Hao Guo, Yan-Pei Cao, and Shi-Min Hu. Charactergen: Efficient 3d character generation from single images with multi-view pose canonicalization. TOG, 2024.
> [2] Jiale Xu, Weihao Cheng, Yiming Gao, Xintao Wang, Shenghua Gao, and Ying Shan. Instantmesh: Efficient 3d mesh generation from a single image with sparse-view large reconstruction models. arXiv preprint, 2024.

---

> > ### Comment · Reviewer_KMF3 · 2024-11-26
> > **Reply to Rebuttal**
> >
> > Thanks for the rebuttal and explanations. The authors' responses clarified some of my previous misunderstandings, and I acknowledge the improvements demonstrated by the authors' results over the previous SOTA works. Therefore, I have raised my score.

---

### Official Review · Reviewer_mvv6 · 2024-11-04

**Soundness:** 3
**Presentation:** 2
**Contribution:** 3
**Rating:** 6
**Confidence:** 5

**Summary:**

The paper introduces a novel method to produce detailed 3D generation with less computational cost and training time. Specifically, the authors propose a sparse differentiable mesh representation called SparseCube to enable more efficient modeling of 3D subjects. Correspondingly, a sparse transformer network is designed to replace tri-plane to accommodate the proposed 3D representation. Through extensive evaluations, the novel 3D representation is demonstrated to be better than DMTet, implicit representation, and voxels. Meanwhile, the experiments showcase the best performance of the proposed method and the effectiveness of each component.

**Strengths:**

+ The paper introduce a novel 3D representation, called SparseCube. It's capable of capturing the details while largely decreasing the computational cost and training time.

+ The proposed method is demonstrated to perform better than existing techniques.

+ Extensive experiments have been provided to demonstrate SparseCube's superiority over DMTet, voxels, and implicit representations.

**Weaknesses:**

- The paper would benefit from further refining the paper structures and polishing the writings.

- The geometry quality in Fig. 1 and Fig. 5 is not good.

- What are the results and comparisons in general 3D avatars like "people wearing daily outfits" or movie characters like "Iron Man"?

- There are no comparisons with human-specified methods like Human-LRM or more recent 3D generative methods like 3DTopia, 3DTopiaXL, etc.

- The design of the features within SparseCube is not well-justified.

**Questions:**

Considering the above strengths and weaknesses, I currently still lean toward a positive rating. I would like to hear from the other reviewers and the authors during the rebuttal.

---

> ### Author Response · Authors · 2024-11-20
> **Rebuttal To Reviewer mvv6**
>
> Thank you for your valuable suggestions and comments! We have tried to address your concerns in the rebuttal text :
>
> **Paper Structures and Writings.**
>
> Thank you for your advice. We have addressed and corrected the issues you mentioned.
>
> **Geometry Quality.**
>
> Generating meshes with extremely high geometric quality is a challenging task. However, compared to other methods, our results already demonstrate significantly higher quality and are the closest to the ground truth. We plan to further enhance the geometry quality in future work.
>
> **Comparison with Human-Specified Methods.**
>
> We have compared our method with Tech [1], a human-specified method. The specific results are shown in Figure 9 in the appendix. From these results, we can observe that SMPL-based methods are not well-suited for anime characters.
>
> **Design of The Features Within SparseCube**
>
> We use learnable positional embeddings in both stages of our method. The positional embedding shapes for the first and second stages are $32^3 \times 1024$ and $64^3 \times 1024$, respectively. During training, the second-stage positional embeddings are initialized by upsampling the first-stage positional embeddings. Additionally, in the second stage, the positional embeddings are sparsified based on the results from the first stage. In the cross-attention mechanism, the positional embeddings act as the query sequence, while the image tokens from DINO serve as the key-value sequence.
>
> [1] Yangyi Huang, Hongwei Yi, Yuliang Xiu, Tingting Liao, Jiaxiang Tang, Deng Cai, and Justus Thies. TeCH: Text-guided Reconstruction of Lifelike Clothed Humans. In 3DV, 2024.

---

### Meta-Review · Area_Chair_B6ri · 2024-12-21

**Metareview:**

The paper introduces a novel method for detailed 3D generation with reduced computational cost and training time. It received two borderline acceptances and one borderline reject. Most reviewers recognize the technical contribution of the proposed SparseCubes and Sparse Transformers.

As noted by Reviewer mvv6, the writing of the paper should be significantly improved. Additionally, Reviewer KMF3 pointed out the lack of comparison with state-of-the-art methods. The rebuttal provides a comparison with CharacterGen and includes more technical details, addressing this concern.

The area chair believes that the writing and the presentation of the results should be further refined. Additionally, the paper references “Instant3D,” which could be easily confused with another relevant work, Instant3D: Instant Text-to-3D Generation (IJCV 2024). This distinction should be clarified in the final version of the paper. Nevertheless, the area chair recognizes the technical contribution and novelty of the work and recommends acceptance.

**Additional Comments On Reviewer Discussion:**

The rebuttal has effectively addressed several key concerns raised by the reviewers, including providing additional technical details and presenting more comparisons with state-of-the-art methods, such as CharacterGen. Reviewer ozD8 raised further questions regarding how the proposed method compares to other sparse-based representations, and the authors have provided additional analysis on the feasibility of using XCUBE’s backbone. Other reviewers did not raise any further concerns after the rebuttal.

---

### Decision · Program_Chairs · 2025-01-22

Accept (Poster)